# Topography and Nanomechanics of the Tomato Brown Rugose Fruit Virus Suggest a Fragmentation-Driven Infection Mechanism

**DOI:** 10.3390/v17091160

**Published:** 2025-08-25

**Authors:** Péter Puskás, Katalin Salánki, Levente Herényi, Tamás Hegedűs, Miklós Kellermayer

**Affiliations:** 1Department of Biophysics and Radiation Biology, Semmelweis University, H-1094 Budapest, Hungary; ekitep059@gmail.com (P.P.); herenyi.levente@semmelweis.hu (L.H.); tamas@hegelab.org (T.H.); 2Department of Plant Pathology, Plant Protection Institute, Centre for Agricultural Research, Hungarian Research Network (HUN-REN), H-1525 Budapest, Hungary; salanki.katalin@atk.hun-ren.hu; 3Hungarian Research Network (HUN-REN) Biophysical Virology Research Group, Semmelweis University, H-1094 Budapest, Hungary

**Keywords:** atomic force microscopy, force map, force spectroscopy, AlphaFold, tomato brown rugose fruit virus

## Abstract

Tomato brown rugose fruit virus (ToBRFV) has been causing severe agricultural damage worldwide since its recent discovery. While related to tobacco mosaic virus, its properties and infection mechanisms are poorly understood. To uncover their structure and nanomechanics, we carried out atomic force microscopy (AFM) measurements on individual ToBRFV particles. The virions are rod-shaped with a height and width of 9 and 30 nm, respectively. Length is widely distributed (5–1000 nm), with a mode at 30 nm. ToBRFV rods displayed a 22.4 nm axial periodicity related to structural units. Force spectroscopy revealed a Young’s modulus of 8.7 MPa, a spring constant of 0.25 N/m, and a rupture force of 1.7 nN. In the force curves a step was seen at a height of 3.3 nm, which is related to virion wall thickness. Wall thickness was also estimated by predicting coat protein structure with AlphaFold, yielding a protein with a length of 7.3 nm. Accordingly, the structural element of ToBRFv is a right circular cylinder with an equal height and diameter of ~22 nm and a wall thickness between 3.3 and 7.3 nm. Thus, at least four to nine serially linked units are required to encapsidate a single, helically organized RNA genome. Fragmentation of ToBRFV into these cylindrical structural units may result in a facilitated release of the genome and thus efficient infection.

## 1. Introduction

Viruses are structurally diverse nanocapsules carrying genetic material that infect host organisms with an array of molecular tricks. The tomato brown rugose fruit virus (ToBRFV) is a positive-sense, single-stranded RNA plant virus classified in the genus *Tobamovirus* (family *Virgaviridae*). It was first recognized as a distinct Tobamovirus species in 2016 [1]. The spreading of ToBRFV has caused major agricultural damage globally [1,2,3,4,5,6]. ToBRFV does not have any known insect vectors and is transmitted mechanically (via contact or contaminated tools/seeds), just like other tobamoviruses. ToBRFV virions are non-enveloped, rigid helical rods consisting of many coat protein subunits encapsidating the RNA. A recent transmission electron microscope study measured the dimensions of ToBRFV particles as ~274.8 nm by 13.9 nm [7], which are comparable to the 300 × 18 nm dimensions reported for related tobamoviruses. The capsid has helical symmetry, with each coat protein (CP) subunit bound to the genomic RNA. Similarly to the tobacco mosaic virus (TMV), ~2130 copies of the CP likely assemble per virion, forming a hollow, helical tube around the RNA. The RNA is tightly intercalated along the inner surface of the protein helix. This “crinkled cylindrical” capsid architecture protects the RNA and gives tobamoviruses their characteristic stability. The CP of ToBRFV comprises 159 amino acids (≈17.5 kDa), very similar to other tobamovirus CPs [8], and it mediates particle assembly and aids long-distance movement in plants. ToBRFV contains a single-stranded, positive-sense RNA genome 6.392 kb in length, with a 5′ methylated cap and a 3′ tRNA-like structure instead of a poly(A) tail. The genome encodes four open reading frames (ORFs). At the 5′ end, ORF1 and ORF2 overlap via a leaky stop codon and together code for the viral replicase, a 126 kDa protein and a 183 kDa readthrough protein that function as subunits of the RNA-dependent RNA polymerase (RdRP) [8,9]. Further downstream, ORF3 encodes the movement protein (MP) (~28–31 kDa), which enables cell-to-cell spreading through plasmodesmata [8,9]. The final ORF4 encodes the CP (~17.5 kDa) that forms the capsid [8,9]. ORF3 and ORF4 are expressed from subgenomic RNAs produced during replication [8,9]. The genomic RNA also has untranslated regions (UTRs) that are important for replication and encapsidation and, notably, a tRNA-like structure at the 3′ end, as in other tobamoviruses. ToBRFV isolates are very similar genetically worldwide (over 99% identity) [10], but if any structural variations exist, they could affect properties such as virulence. It remains unknown whether any mutations in the coat protein or other structural regions of ToBRFV have functional consequences on stability or host interactions. So far, the main resistance-breaking determinant was found in the movement protein [8], not in the CP, and serologically the CP is cross-reactive with TMV and tomato mosaic virus (ToMV). This suggests that the CP structure is highly conserved. We currently lack detailed knowledge of how ToBRFV disassembles inside the host. In TMV, the process by which the coat protein is stripped off the RNA is only partly understood, and for ToBRFV it is entirely inferred. A recent cryo-electron microscopy study pointed out that changes in the local pH and calcium levels may modulate the interaction between key glutamates (called Caspar-carboxylates [11]) of neighboring CPs, serving as a disassembly switch in TMV [12]. Whether a similar mechanism may operate in ToBRFV and whether structural intermediates are present during disassembly are not known.

Here we investigated the topographical structure and nanomechanical properties of ToBRFV by using atomic force microscopy and force mapping. We found that ToBRF virions are soft rodlike structures that can be fragmented into unit-length elements which may have important implications for infection efficiency.

## 2. Materials and Methods

### 2.1. Isolation of the ToBRF Virus

The investigated ToBRFV isolate was collected in a tomato fruit (*Solanum lycopersicum*)-producing greenhouse in Lébény, Hungary, in May 2022. This location is the same as the one from where the first Hungarian appearance of ToBRFV (genus *Tobamovirus*) was reported in 2021 [13]. Tomato leaf samples with typical mosaic symptoms were collected. The presence of ToBRFV was verified with RT/PCR and nucleotide sequence determination (GenBank PV324971), as previously reported [14]. The virus was propagated by mechanical inoculation into tomato plants (*Solanum lycopersicum* cv Moneymaker). Inoculated plants were kept in a growth chamber at 24 °C and exposed to a 14/10 (h/h) light/dark cycle. Virus particles were purified by polyethylene glycol precipitation [15]). Then, 100 µL aliquots of the virion suspension were frozen in liquid N_2_ and stored at −80 °C until further use.

### 2.2. Preparation of ToBRFV Samples for AFM

ToBRFV particles were imaged with AFM either in air or in buffer solution. For scanning in air, a 20 μL aliquot of purified ToBRFV sample, diluted appropriately in phosphate-buffered saline (PBS; 10 mM K-phosphate, pH 7.2, 140 mM NaCl, 0.02% NaN_3_), was pipetted onto freshly cleaved mica and incubated for one minute at room temperature. Subsequently, the surface was rinsed with milliQ water (Merck, Darmstadt, Germany) to remove unbound virions, and the sample surface was dried with a gentle stream of high-purity nitrogen gas. When scanning under aqueous buffer conditions, the unbound virions were removed by washing with PBS. After the last wash, 100 µL of PBS was left on the sample, and the imaging was carried out under aqueous buffer conditions.

### 2.3. AFM Imaging

ToBRFV particles were imaged with AFM using procedures employed before [16,17,18,19,20]. Briefly, imaging was carried out in AC (tapping, non-contact) mode using either an Asylum Research MFP3D or a Cypher ES instrument (Oxford Instruments, Santa Barbara, CA, USA). All measurements were performed at room temperature (25 °C). When imaging in air or buffer, we used AC160TS or BL-AC40TS cantilevers (Olympus Corporation, Tokyo, Japan), respectively. The cantilevers were oscillated at or near their resonance frequencies (~290 kHz for AC160TS and ~20 kHzBL-AC40TS) using piezo or photothermal excitation. Typically, 512 × 512-pixel scans were obtained at line-scan rates of 0.7–1.5 Hz.

### 2.4. Force Mapping and Spectroscopy

Individual ToBRFV particles were mechanically manipulated by methods used before [17,19,20]. Briefly, the ToBRFV-coated surface was manipulated by measuring force spectra in indentation–retraction cycles at each pixel (typically 64 × 64 pixel map). BL-AC40TS (Olympus Corporation, Tokyo, Japan) cantilevers were used, the stiffness of which was calibrated with the thermal method. The typical maximum indentation force was 4 nN. Cantilever movement rate varied between 0.5 and 1.5 µm/s. Force data were acquired at a sampling rate of 1 kHz.

### 2.5. Image Processing, Data Analysis, Calculations, and Statistics

Post-processing and analysis of AFM images were conducted semi-automatically using the AsylumResearch AFM driver software (v.16.00148) (running under IgorPro v.6.3, Wavemetrics, Lake Oswego, OR, USA). The process began with flattening the images to eliminate artificial distortions and ensure a planar background. Particles were identified by masking with a threshold at the average half-maximal topographical height. Particle analysis was carried out to obtain the maximum topographical height, length, width, volume, and circularity (isoperimetric quotient) of the ToBRFV virions. In excess of 1600 ToBRFV virions were analysed.

The force map data were used to calculate structural (virion height and wall thickness) and mechanical (Young’s modulus, stiffness, and rupture force) parameters of ToBRFV virions. Based on the pixel-to-pixel force (*F*) data of the force map images, we calculated the spatial distribution of the Young’s modulus (*E*) using the Hertz model: (1)$$E=\frac{F\pi \left(1-\upsilon^{2}\right)\tan\theta }{2d^{2}},$$
where *ν* is the Poisson ratio (0.33), *θ* is the angle between the horizontal plane and the side wall of the conical AFM tip (Olympus AC160TS, cone angle: 36°), and *d* is the indentation depth. Capsid stiffness (*κ*) was calculated from the slope of the force-versus-distance (*s*) curves within the pixels representing ToBRFV particles identified in the force maps as(2)$$\kappa =\frac{\Delta F}{\Delta s}$$

The perimeter (*P*) of the elliptical virion cross section was calculated using Ramanujan estimation [21] as(3)$$P\approx \pi \left[3\left(a+b\right)-\sqrt{\left(3a+b\right)\left(a+3b\right)}\right],$$
where *a* and *b* are the major and minor axes of the ellipse, respectively. From the calculated perimeter we obtained the diameter (*D*) of the cylindrical ToBRFV particles as(4)$$D=\frac{P}{\pi }$$

Data were analyzed, compared, and plotted using commercial software (MS Excel, IgorPro versions 6 and 8).

### 2.6. Structural Analysis

The structure of the ToBRFV cap protein was predicted using AlphaFold, using the amino acid sequence of the 17.5 kDa protein (GenBank: WJN56905.1, Appendix A). AlphaFold3 and its associated databases were installed locally, following the instructions at https://github.com/google-deepmind/alphafold3 (accessed on 10 March 2025), and configured to run on a system with an AMD 16-core processor and an NVIDIA RTX A6000 GPU with 48 GB of VRAM. The PyMOL Molecular Graphics System (Version 2.4, Schrödinger, LLC, New York, NY, USA) was used for analyzing and visualizing the predicted structures.

## 3. Results

### 3.1. ToBRFV Virions Are Rodlike Particles with Wide Length Distribution

Tomato brown rugose fruit virus (ToBRFV) virions adsorbed to a mica surface appeared as rodlike particles (Figure 1a). A pronounced feature was that while the width of the particles was uniform, their length varied greatly. Larger-magnification AFM images (Figure 1b) confirmed the rodlike shape of ToBRFV virions. Although most of the virion rods appeared straight, some of them displayed a slight curvature. The width of the rods appeared uniform, and their ends were rounded. Interestingly, most of the virions displayed cross-striations, seen more clearly in the amplitude-contrast images (Figure 1b, right). Some of the virions appeared spherical, indicating that the length of the rods was equal to their width.

To extract structural features of the ToBRFV virions, we carried out particle analysis (Figure 1c–f and Appendix A). The topographical height of the virions, measured as the height from the substrate surface in the particle center, had a peak at 9 nm (Figure 1c and Appendix A). The length of the virions was distributed across a very wide range (5–1000 nm) and displayed a log-normal distribution with a mode at 30 nm (Figure 1d and Appendix A). Local modes appeared at longer lengths in the histogram (Appendix A), suggesting that the ToBRFV virions might be a serial multiple of a unit-length elementary structure. Occasionally we observed viruses with lengths far exceeding the range (Appendix A), pointing to the capacity of ToBRFV for unhindered axial assembly. The virion width, measured at the half-maximal topographical height, was 28.4 nm (±8.2 nm SD) with a mode at 30 nm (Figure 1e and Appendix A). Notably, the modal width is more than three times as large as the modal virion height, indicating that the virus particles flattened upon binding to the mica surface. The circularity of the ToBRFV AFM image particles (Figure 1f) ranged from 0.05 to 1.0, which corresponds to the wide shape distribution from long rods to circles, respectively. The histogram indicates that circular particles dominated the sample. Thus, the ToBRFV virus particles are flattened rodlike structures with widely ranging lengths that indicate a propensity for fragmentation.

### 3.2. Virions Display an Axially Periodic Structure

We analyzed the axial topography of individual ToBRFV particles in greater detail (Figure 2). Some virions displayed a cleft along the axial contour, such that the virion appeared to be on the verge of fragmenting (Figure 2a). Along the surface of most of the virions, shallow ridges could be observed. The profile plots along the virion axis revealed periodic variation in topographical height, the peak-to-peak value of which reached up to 0.8 nm, which is nearly 10% of the modal virion height. The periodicity of the height variations, measured as the axial distance between consecutive peaks, was 22.4 nm (±8.5 SD) (Figure 2b). It is quite plausible that the ridges correspond to the boundaries across which virion fragmentation, seen in overview AFM images (Figure 1a), occurs. Altogether, the topography of the ToBRFV particles is not smooth but displays ridges which may be sites of fragmentation into structural building blocks.

### 3.3. ToBRFV Virions Are Soft and Fragile

We investigated the nanomechanical behavior of ToBRF virions under aqueous buffer conditions by employing force mapping (Figure 3). In this experimental approach each pixel of the acquired images contains a force-versus-distance function collected in repeated indentation–retraction cycles. Prior to measuring the force map, we carried out AC-mode AFM imaging of the sample (Figure 3a). Based on the force map, we calculated the spatial distribution of the Young’s modulus (Figure 3b) based on fitting with the Hertz model (Equation (1), Appendix A). The ToBRFV Young’s modulus histogram, calculated by masking the rigid background, shows a peak centered at 8.7 MPa (±4.7 SD) (Figure 3c), which is nearly three orders of magnitude smaller than that measured for the TMV earlier [22].

Considering that each pixel of the force map contains an entire force–distance function, we investigated these force traces to uncover further structural and mechanical information about ToBRFV. In a typical indentation–retraction curve (Figure 3d), we identified different features and force-driven structural transitions. First, we could identify the instant at which the cantilever tip landed on the virion, which allowed us to measure the topographical height of the capsid. Second, we observed a linear elastic deformation, which allowed us to measure capsid stiffness. Third, we observed a force peak, which corresponds to capsid rupture. Measuring the force associated with the peak allowed us to identify the rupture force, which is a measure of viral mechanical stability [20,23]. Finally, we often observed a residual thickness following the capsid rupture force peak, which may be assigned to the thickness of the back wall of the virion directly in contact with the substrate surface. Measuring this height allowed us to estimate the ToBRFV wall thickness. The virion height histogram (Figure 3e), obtained from the indentation force traces, displayed a Gaussian distribution centered at 10.4 nm (±1.2 SD), which is nearly 15% greater than that measured by particle analysis (Figure 1c). Considering that the force mapping experiments were carried out under aqueous buffer conditions, in contrast to the dry image acquisition for particle analysis, our results suggest that water plays a role in the structure of ToBRFV. The stiffness results (Figure 3f) showed a mean of 0.25 N/m (±0.12 SD), which is threefold smaller than that measured for T7 phages before [20]. The mean capsid rupture force (Figure 3g) was 1.7 nN (±0.66 SD), which is about one-fourth of that measured earlier for the T7 phage [20] or TMV [24]. The wall thickness histogram (Figure 3h) indicated a mean value of 3.3 nm (±0.8 SD). Notably, in some of the force spectra, force peaks and non-linear force traces appeared in the retraction phase of the mechanical cycle (Appendix A), indicating that genomic RNA could be pulled. A height-contrast adjustment test showed that the nanoindentation experiments indeed ruptured the ToBRFV capsids rather than dislodged them from the surface (Appendix A). Altogether, the ToBRFV capsid appears softer and mechanically less stable than numerous icosahedral and other helical viruses investigated earlier.

## 4. Discussion

In the present work we investigated the topographical structure and nanomechanical properties of the tomato brown rugose fruit virus (ToBRFV). ToBRFV, a member of the *Tobamovirus* genus, is a recently emerged virus, the spread of which has caused major agricultural damage globally [1,2,3,4,5,6]. Understanding the detailed features of ToBRFV is important in developing preventive and defensive measures. Furthermore, the structural and mechanical information collected may help us better understand the behavior of further pathogenic tobamoviruses.

ToBRFV appeared in AFM images (Figure 1) as rodlike particles with a remarkably wide length distribution (5–1000 nm), and occasionally even longer viruses could be observed (Appendix A). The rods displayed rounded ends. However, the rounded ends might be the result of convolution between the cylindrical virion and the conical AFM tip. Therefore, at present we are cautious about the interpretation, as the ToBRFV virions are likely to be open-ended cylinders, similar to TMV [25]. Indeed, recent electron microscopic analysis of ToBRFV particles indicated that they are blunt-ended [1]. The length histogram displayed a mode at 30 nm, indicating that a ~30 nm long element may be a structural unit of the virus, from which it assembles or into which it disassembles. We point out, however, that the particle analysis always assigns the long axis of a particle image to its length, but this long axis might not coincide with the long axis of the virus. In other words, if a ToBRFV fragment is shorter than its width, then the width value is assigned to the particle length because this is the longer value. These short fragments will then increase the population of particles with a length equal to the virion width. Altogether, the length of an elementary structural segment of ToBRFV is smaller than or equal to ~30 nm.

The topographical height of the ToBRFV particles was remarkably (more than threefold) smaller than their width, indicating that they spread across the substrate surface. This behavior reflects the capacity for deformation; hence, the ToBRFV virions appear to be rather compliant. The virion height-versus-length function (Appendix A) indicates that the peak height (9 nm) is characteristic of virions with a length between 30 and 60 nm. Below 30 nm the height drops sharply, which is most likely due to breakage of the cylindrical structure altogether. Above 60 nm the height decreases monotonically by ~0.5 nm in virions several hundred nanometers long. We speculate that the effect may be related to the length-dependence of bending rigidity. From the width and height data, one may compute the cross-sectional perimeter, assuming that it is likely to be a flattened ellipse (Appendix A). The isotropic (i.e., equivalent distance units along the x- and y-axes) representation of the cross-sectional height profile (Appendix A) indicates that the width measured at half-maximal topographical height corresponds to the real virion width. Accordingly, the height (Figure 1c) and width (Figure 1e) data correspond to the short and long axes of the cross-sectional ellipse. From these data we approximated the perimeter (Equation (3)), which allowed us to calculate the diameter (Equation (4)) of the equilibrium cylindrical ToBRFV virion as 22 nm.

High-resolution AFM images revealed a 22.4 nm (±8.5 SD) periodicity along the long axis of the ToBRFV virions (Figure 2). We hypothesize that the uncovered periodic transverse ridges reflect boundaries of elementary structural units of ToBRFV from which it is constructed during viral assembly or into which it fragments during infection. Accordingly, the length of the elementary structural unit is ~22 nm. Thus, the unit-length structural element of ToBRFV is a right circular cylinder, the height and diameter of which are identical. Conceivably, this unit-length element represents a highly stable configuration; were its length significantly higher or lower than its diameter, it would be easily deformed by axial or tangential bending forces, respectively.

Nanomechanical manipulation of surface-adsorbed ToBRF virions indicated that they are softer and mechanically less stable than numerous icosahedral or helical viruses investigated so far [20,22]. The mechanical stability of viruses is intimately tied to and balanced with their survival, host interactions and entry, and the release of their genome [26,27]. We suspect that the relatively low mechanical stability of ToBRFV observed here may facilitate the efficient genome release. Interestingly, the height of the ToBRF virions was nearly 15% greater when measured with nanoindentation (wet conditions) than with particle analysis (dry conditions). We hypothesize that water, via hydration of the coat protein and RNA, might play a role in the structure and probably the mechanical behavior of ToBRFV. In support, water has been shown to affect the nanoscale mechanical properties of TMV [28]. Although the exact mechanisms of how water influences the properties of viral molecular components need to be investigated further, the water-dependent structural and mechanical features likely play a role in the stability, survival, and transmission of viruses [29,30]. The nanoindentation force spectra revealed that following capsid rupture a compressible structure remains, which we assigned to the back wall of the virion. The mean wall thickness was 3.3 nm (±0.8 SD), which is significantly smaller than that of TMV [25]. Since the measured thickness corresponds to that of the ruptured, broken capsid, it is likely that the structure of the wall has been compromised. Nevertheless, this value provides a lowermost estimate of the capsid wall thickness. Notably, in some of the force spectra, force peaks and non-linear force traces appeared in the retraction phase of the mechanical cycle (Appendix A). The observation indicates that filamentous structures were pulled out of the broken capsid, and the applied mechanical force induced conformational changes resulting in the release of further filament segments. Considering that the length of these filaments often reached beyond 250 nm, which far exceeds the unfolded length of a single coat protein (~60 nm), the filaments likely correspond to the genomic RNA of ToBRFV. Because the force peaks (sometimes reaching several hundred piconewtons) were much greater than the forces required to unfold RNA hairpins (~15 pN) [31], they likely reflect the mechanically driven dissociation of RNA from the coat proteins constituting the capsid. The results suggest that the genomic RNA of ToBRFV is tightly associated with the capsid wall, hence the coat proteins.

The structural and nanomechanical information gathered in our experiments allows us to propose a model of the ToBRF virion (Figure 4). Given that the nanoindentation experiments provided only a lowermost estimate of the capsid wall, we carried out a structural prediction of the 17.5 kDa coat protein (CP) of ToBRFV using AlphaFold (Figure 4a,b and Appendix A) so that the wall thickness could be more precisely estimated. Based on the amino acid sequence of CP (Appendix A), AlphaFold predicted a protein dominated by alpha-helices with short, disordered regions in the middle and at the C-terminus (Figure 4a). The length of the protein from the C-terminal SER159 to ALA101, located in a disordered loop at the other end of the molecule, is 7.3 nm. This length provides an uppermost estimate of the capsid wall thickness, as the exact orientation of the CP within the wall is not known. However, predictions involving multiple CP molecules (Figure 4b and Appendix A) strongly suggest that this upper estimate represents a plausible in situ thickness. The lower- and uppermost wall thickness estimates form the basis of two models (a and b) (Appendix A), the parameters of which are listed in Table 1. We hypothesize that the genomic ssRNA is helically organized within the core of the cylindrical capsid, similarly to TMV [25]. Importantly, the capsid’s inner diameter far exceeds the persistence length of ssRNA (assumed to be identical to ssDNA, ~0.6 nm [32]) in either model; therefore, in contrast to bacteriophages [33], energy need not be invested in the incorporation of the genome. On the contrary, the genomic RNA, aided by its entropic elasticity, likely assists in holding the capsid together. Based on the width of the CP (Figure 4a), we assume that the pitch of the genomic ssRNA helix is 2 nm. Accordingly, the ToBRFV genome (6392 bases) extends across several unit-length cylindrical elements in both models (four and nine). Considering that an ~200 nm average length has been reported recently for ToBRF virions by electron microscopy [1], model b is likely to more closely reflect the true structural arrangement. We note, however, that the coat proteins must re-arrange considerably to give way to the significant flattening when binding to the substrate surface (Appendix A). Regardless of the model, the unit-length elements and the fragility of ToBRFV may have important implications for the infection mechanism. Conceivably, during infection the virions become exposed to bending and shear forces, leading to fragmentation (Figure 4c). Because the genome spans numerous unit-length elements, the fragmentation results in the exposure of the genome at several locations at the same time, which is a much more efficient genome release mechanism than diffusion out of a long cylinder. Whether the unit-length cylindrical units play a role during virus assembly remains to be investigated.

## 5. Conclusions

In conclusion, ToBRFV virions are rodlike particles that can be fragmented into unit-length structures. The unit-length element is a right circular cylinder with equal height and diameter measurements (~22 nm), which likely represents a mechanically optimal structure, allowing concatenation to lengths beyond a micrometer. Fragmentation of the virion permits the exposure of its genome at multiple sites, thereby increasing the efficiency of infection.

## Figures and Tables

**Figure 1 viruses-17-01160-f001:**
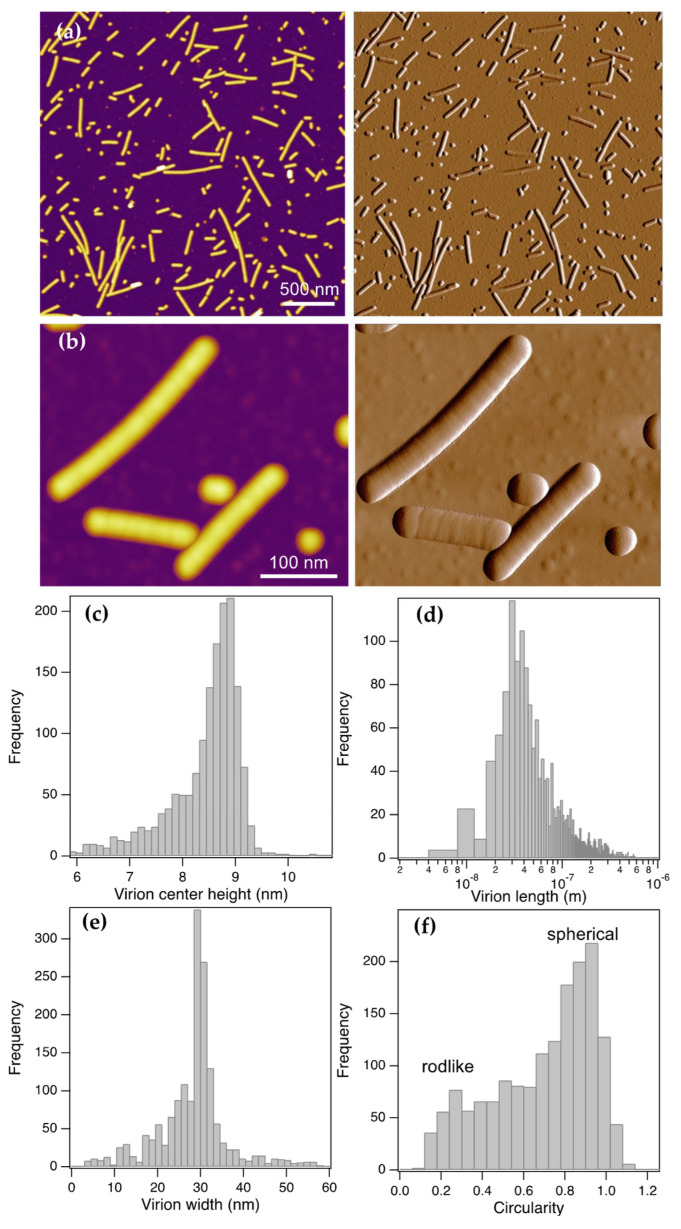
AFM imaging and particle analysis of surface-adsorbed ToBRFV virions under dry, ambient conditions. Height (**left**) and amplitude (**right**) AFM images with a large field of view (**a**) and a small area of the sample (**b**). Distribution of the center height (**c**), length (**d**), width (**e)** and circularity (**f**) of ToBRFV particles. Particles were segregated by masking at the half-maximal height of the particles.

**Figure 2 viruses-17-01160-f002:**
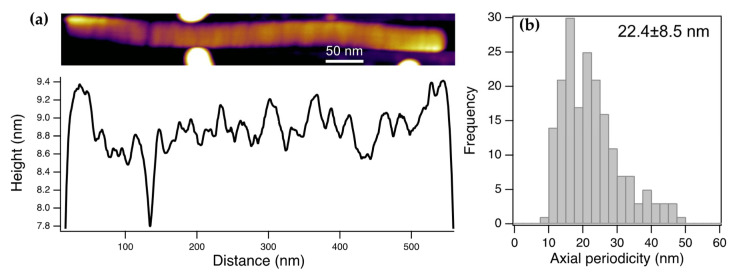
Axial topographical height variations in ToBRFV virions. (**a**) Height-contrast AFM image (scanned dry) of a long ToBRFV particle displaying a ridge and axial periodicity (**top**) and an axial profile plot along the length of the virus particle (**bottom**). (**b**) Distribution of axial periodicity, measured as the distance between consecutive topographical peaks.

**Figure 3 viruses-17-01160-f003:**
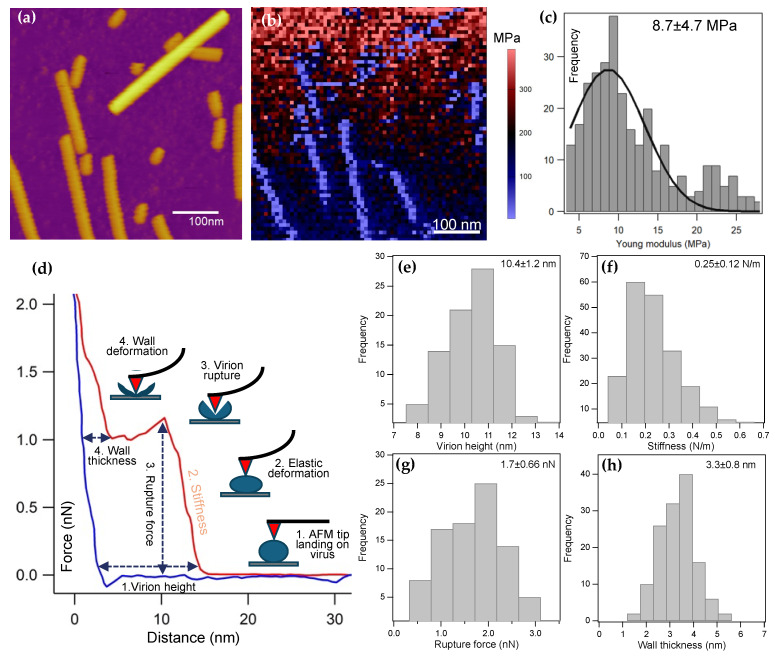
Force map and spectroscopy analysis of surface-adsorbed ToBRFV particles. (**a**) Height-contrast AFM image collected after 30 min of sample incubation on substrate surface. (**b**) Young’s modulus map of the sample shown in (**a**). (**c**) Distribution of Young’s modulus values obtained from the pixels containing ToBRFV mechanical information. (**d**) Force-versus-distance curve collected from a ToBRFV virion. Red and blue traces correspond to the indentation and retraction phases of the mechanical cycle, respectively. The explanatory inset figures explain the stages of the experiment which yield particular structural and mechanical information. Distributions of virion height (**e**), stiffness (**f**), rupture force (**g**), and wall thickness (**h**).

**Figure 4 viruses-17-01160-f004:**
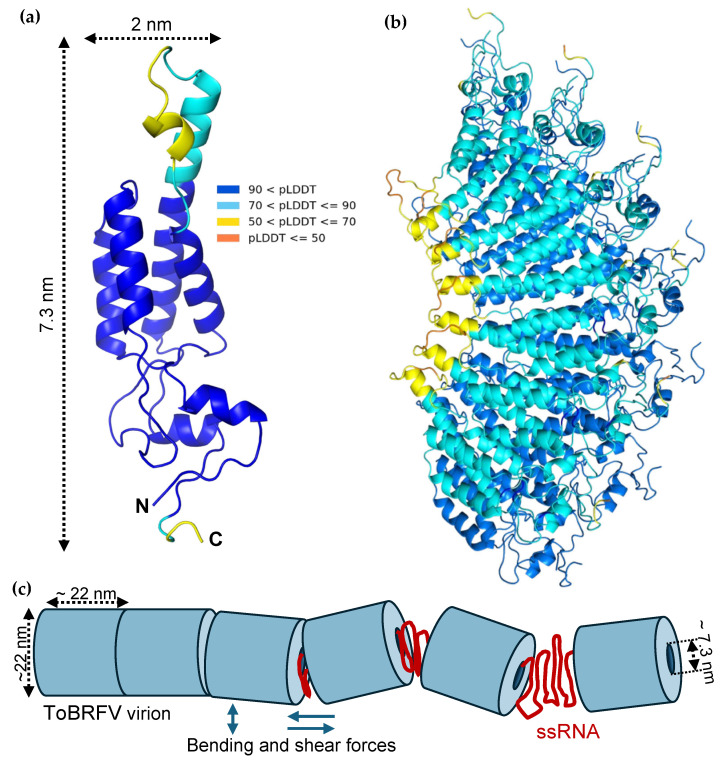
Modeling the ToBRFV coat protein and virion. (**a**) Structural model of the 17.5 kDa ToBRFV coat protein (CP), predicted by AlphaFold3 based on the amino acid sequence (see Appendix A). The colors represent the predicted reliability of the structure, with higher pLDDT values indicating greater confidence. (**b**) Thirteen CP protomers are arranged in a partial disk, forming two layers. One layer is colored by pLDDT to indicate confidence, while the other is shown in blue for clarity. Structures and predicted aligned error (PAE) plots for models with varying numbers of protomers are shown in Appendix A. (**c**) Schematic model of ToBRFV fragmentation and ssRNA release upon exposure to mechanical (bending and shear) forces during infection.

**Table 1 viruses-17-01160-t001:** Structural parameters of the ToBRFV model virions. The two models are shown in Appendix A. In model a, we used the presumed virion wall thickness obtained from nanomechanical data (see Figure 3). In model b, we used the dimensions of the coat protein, the structure of which was predicted with AlphaFold (Figure 4a).

Parameter	Model a	Model b
Outer diameter	22 nm	22 nm
Wall thickness	3.3 nm	7.3 nm
Inner diameter	15.4 nm	7.4 nm
Inner perimeter	48.4 nm	23.2 nm
Number of RNA bases per turn (0.34 nm/base)	142	68
Total RNA turns per genome (6392 bases)	45	94
Virion length (2 nm helical pitch)	90 nm	188 nm
Number of unit-length elements in virion	4	9

## Data Availability

The original contributions presented in this study are included in the article/Appendix A. Further inquiries can be directed to the corresponding author.

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
