# Peer review of "Topography and Nanomechanics of the Tomato Brown Rugose Fruit Virus Suggest a Fragmentation-Driven Infection Mechanism"

_viruses, 2025, doi:10.3390/v17091160_

Round 1
Reviewer 1 Report
Comments and Suggestions for Authors
This manuscript study presents a systematic analysis of the morphology and nanomechanical properties of ToBRFV using atomic force microscopy and force spectroscopy, integrated with AlphaFold structural predictions. The results reveal a potential fragmentation-driven infection mechanism. The experimental design is rigorous and the data comprehensive, providing a novel perspective on the structure and infection mechanisms of ToBRFV, with significant scientific implications and application potential. I have few minor concerns for authors to improve as below.
Line 320 ("We suspect that water..."): The proposed role of water remains speculative, as no supporting experimental evidence is provided to substantiate this hypothesis.
Regarding mechanical comparisons with TMV (Lines 255 & 317): While ToBRFV virions are described as "softer and more fragile" than TMV, the biological implications of these mechanical differences, particularly regarding infection efficiency or host, remain unexplored. I recommend expanding the Discussion to address this relationship.
Author Response
Response to Reviewer 1
Comment 1:
This manuscript study presents a systematic analysis of the morphology and nanomechanical properties of ToBRFV using atomic force microscopy and force spectroscopy, integrated with AlphaFold structural predictions. The results reveal a potential fragmentation-driven infection mechanism. The experimental design is rigorous and the data comprehensive, providing a novel perspective on the structure and infection mechanisms of ToBRFV, with significant scientific implications and application potential. I have few minor concerns for authors to improve as below.
Response:
We thank the reviewer for the overall highly positive and supportive opinion and comments. The comments were duly addressed in our revision.
Comment 2:
Line 320 ("We suspect that water..."): The proposed role of water remains speculative, as no supporting experimental evidence is provided to substantiate this hypothesis.
Response:
Thank you for this critical remark. We agree that mentioning the role of water is somewhat speculative in this stage of the experimentation. However, considering the interesting nature of the observation and the generally important role of water in plants, we would opt for keeping the notion but amending it with cautious explanation and references (lines 323-329 of the revised manuscript).
Comment 3:
Regarding mechanical comparisons with TMV (Lines 255 & 317): While ToBRFV virions are described as "softer and more fragile" than TMV, the biological implications of these mechanical differences, particularly regarding infection efficiency or host, remain unexplored. I recommend expanding the Discussion to address this relationship.
Response:
Thank you for the comment. In the revised manuscript the Discussion was complemented with a brief notion about the possible role of mechanics in viral infectivity (lines 318-321 of the revised manuscript).
Reviewer 2 Report
Comments and Suggestions for Authors
The manuscript presents interesting and thoroughly conducted research on the structure and nanomechanical properties of Tomato Brown Rugose Fruit Virus (ToBRFV), using atomic force microscopy (AFM) and AlphaFold-based structural modeling. The authors propose a new hypothesis, according to which fragmentation of virions into cylindrical structural units promotes genome release and increases infection efficiency. The work is timely and relevant in the context of global losses in tomato crops. The manuscript is well-written, the methods used are appropriate, and the illustrations are clear. Figure 3d is especially beneficial in illustrating the mechanical stages of virion indentation. The authors provide new information regarding the mechanical fragility and structural periodicity of ToBRFV virions. The hypothesis regarding the fragmentation-based infection mechanism is original and well-documented. Figures are informative and well-labeled. However, several issues require clarification or expansion. After considering the comments below, I recommend acceptance for publication.
Minor comments
- The discussion on the role of hydration in virion structure is insightful. It would be beneficial to elaborate on how this might affect virion stability in planta.
- The comparison with TMV and T7 phage is valid; however, further discussion on how these differences might translate into biological behavior (e.g., host range, transmission) would enrich the manuscript.
- Typo in line 341: “sructural” should be “structural”.
- Line 280: “ponting” should be “pointing”.
- Please ensure consistent formatting of units (e.g., nm, MPa) throughout the manuscript.
Author Response
Response to Reviewer 2
Comment 1:
The manuscript presents interesting and thoroughly conducted research on the structure and nanomechanical properties of Tomato Brown Rugose Fruit Virus (ToBRFV), using atomic force microscopy (AFM) and AlphaFold-based structural modeling. The authors propose a new hypothesis, according to which fragmentation of virions into cylindrical structural units promotes genome release and increases infection efficiency. The work is timely and relevant in the context of global losses in tomato crops. The manuscript is well-written, the methods used are appropriate, and the illustrations are clear. Figure 3d is especially beneficial in illustrating the mechanical stages of virion indentation. The authors provide new information regarding the mechanical fragility and structural periodicity of ToBRFV virions. The hypothesis regarding the fragmentation-based infection mechanism is original and well-documented. Figures are informative and well-labeled. However, several issues require clarification or expansion. After considering the comments below, I recommend acceptance for publication.
Response:
We thank the reviewer for the overall highly positive and supportive opinion and comments. The comments were duly addressed in our revision.
Minor comments
- The discussion on the role of hydration in virion structure is insightful. It would be beneficial to elaborate on how this might affect virion stability in planta.
Response:
Thank you for the insightful remark. The revised manuscript was amended with references and a brief and cautious consideration (lines 323-329 of the revised manuscript). - The comparison with TMV and T7 phage is valid; however, further discussion on how these differences might translate into biological behavior (e.g., host range, transmission) would enrich the manuscript.
Response:
Thank you for the positive comment. In response, we amended the manuscript with references and a brief note on the role of mechanical properties on various stages of their life cycle (lines 318-321 of the revised manuscript). - Typo in line 341: “sructural” should be “structural”.
Response:
Thank you for pointing out the error, which is now corrected. - Line 280: “ponting” should be “pointing”.
Response:
Thank you for pointing out the error, which is now corrected. - Please ensure consistent formatting of units (e.g., nm, MPa) throughout the manuscript.
Response:
Thank you for pointing out the issue with units. In the revised manuscript we made sure that the commonly used relevant international units are used.